# Scalable training of artificial neural networks with adaptive sparse connectivity inspired by network science

Decebal Constantin Mocanu [1,2], Elena Mocanu [2,3], Peter Stone[4], Phuong H. Nguyen[2], Madeleine Gibescu[2] & Antonio Liotta [5]

Through the success of deep learning in various domains, artificial neural networks are currently among the most used artificial intelligence methods. Taking inspiration from the network properties of biological neural networks (e.g. sparsity, scale-freeness), we argue that (contrary to general practice) artificial neural networks, too, should not have fully-connected layers. Here we propose sparse evolutionary training of artificial neural networks, an algorithm which evolves an initial sparse topology (Erdős–Rényi random graph) of two consecutive layers of neurons into a scale-free topology, during learning. Our method replaces artificial neural networks fully-connected layers with sparse ones before training, reducing quadratically the number of parameters, with no decrease in accuracy. We demonstrate our claims on restricted Boltzmann machines, multi-layer perceptrons, and convolutional neural networks for unsupervised and supervised learning on 15 datasets. Our approach has the potential to enable artificial neural networks to scale up beyond what is currently possible.

[1] Department of Mathematics and Computer Science, Eindhoven University of Technology, De Rondom 70, 5612 AP Eindhoven, The Netherlands. [2] Department of Electrical Engineering, Eindhoven University of Technology, De Rondom 70, 5612 AP Eindhoven, The Netherlands. [3] Department of Mechanical Engineering, Eindhoven University of Technology, De Rondom 70, 5612 AP Eindhoven, The Netherlands. [4] Department of Computer Science, The University of Texas at Austin, 2317 Speedway, Stop D9500, Austin, TX 78712-1757, USA. [5] Data Science Centre, University of Derby, Lonsdale House, Quaker Way, Derby DE1 3HD, UK. Correspondence and requests for materials should be addressed to D.C.M. (email: d.c.mocanu@tue.nl)

Artificial neural networks (ANNs) are among the most successful artificial intelligence methods nowadays. ANNs have led to major breakthroughs in various domains, such as particle physics[1], deep reinforcement learning[2], speech recognition, computer vision, and so on[3]. Typically, ANNs have layers of fully-connected neurons[3], which contain most of the network parameters (i.e. the weighted connections), leading to a quadratic number of connections with respect to their number of neurons. In turn, the network size is severely limited, due to computational limitations.

By contrast to ANNs, biological neural networks have been demonstrated to have a sparse (rather than dense) topology[4,5], and also hold other important properties that are instrumental to learning efficiency. These have been extensively studied in ref. [6] and include scale-freeness[7] (detailed in Methods section) and small-worldness[8]. Nevertheless, ANNs have not evolved to mimic these topological features[9,10], which is why in practice they lead to extremely large models. Previous studies have demonstrated that, following the training phase, ANN models end up with weights histograms that peak around zero[11–13]. Moreover, in our previous work[14], we observed a similar fact. Yet, in the machine learning state-of-the-art, sparse topological connectivity is pursued only as an aftermath of the training phase[13], which bears benefits only during the inference phase.

In a recent paper, we introduced compleX Boltzmann machines (XBMs), a sparse variant of restricted Boltzmann machines (RBMs), conceived with a sparse scale-free topology[10]. XBMs outperform their fully-connected RBM counterparts and are much faster, both in the training and the inference phases. Yet, being based on a fixed sparsity pattern, XBMs may fail to properly model the data distribution. To overcome this limitation, in this paper, we introduce a sparse evolutionary training (SET) procedure, which takes into consideration data distributions and creates sparse bipartite layers suitable to replace the fully-connected bipartite layers in any type of ANNs.

SET is broadly inspired by the natural simplicity of the evolutionary approaches, which were explored successfully in our previous work on evolutionary function approximation[15]. The same evolutionary approaches have been explored for network connectivity in ref. [16], and for the layers architecture of deep neural networks[17]. Usually, in the biological brain, the evolution processes are split in four levels: phylogenic at generations time scale, ontogenetic at a daily (or yearly) time scale, epigenetic at a seconds to days scale, and inferential at a milliseconds to seconds scale[18]. A classical example which addresses all these levels is NeuroEvolution of Augmenting Topologies (NEAT)[19]. In short, NEAT is an evolutionary algorithm which seeks to optimize both the parameters (weights) and the topology of an ANN for a given task. It starts with small ANNs with few nodes and links, and gradually considers adding new nodes and links to generate more complex structures to the extent that they improve performance. While NEAT has shown some impressive empirical results[20], in practice, NEAT and, most of its direct variants have difficulty scaling due to their very large search space. To the best of our knowledge, they are only capable of solving problems, which are much smaller than the ones currently solved by the state-of-the-art deep learning techniques, e.g. object recognition from raw pixel data of large images. In ref. [21], Miconi has tried to use NEAT like principles (e.g. addition, deletion) in combination with stochastic gradient descent (SGD) to train recurrent neural networks for small problems, due to a still large search space. Very recently in refs. [22,23], it has been shown that evolution strategies and genetic algorithms, respectively, can train successfully ANNs with up to four million parameters as a viable alternative to DQN[2] for reinforcement learning tasks, but they need over 700 CPUs to do so. To avoid being trapped in the same type of scalability issues, in SET, we focus on using the best from both worlds (i.e. traditional neuroevolution and deep learning). E.g., evolution just at the epigenetic scale for connections to yield a sparse adaptive connectivity, structured multi-layer architecture with fixed amounts of layers and neurons to obtain ANN models easily trained by standard training algorithms, e.g. SGD, and so on.

Here, we claim that topological sparsity must be pursued starting with the ANN design phase, which leads to a substantial reduction in connections and, in turn, to memory and computational efficiency. We show how ANNs perform perfectly well with sparsely connected layers. We found that sparsely connected

---

**Box 1 | Sparse evolutionary training (SET) pseudocode is detailed in Algorithm 1**

**Algorithm 1: SET pseudocode**

```
1   %Initialization;
2   initialize ANN model;
3   set ε and ζ;
4   for each bipartite fully-connected (FC) layer of the ANN do
5   |   replace FC with a Sparse Connected (SC) layer having a Erdős-Rényi topology given by ε and Eq.1;
6   end
7   initialize training algorithm parameters;
8   %Training;
9   for each training epoch e do
10  |   perform standard training procedure;
11  |   perform weights update;
12  |   for each bipartite SC layer of the ANN do
13  |   |   remove a fraction ζ of the smallest positive weights;
14  |   |   remove a fraction ζ of the largest negative weights;
15  |   |   if e is not the last training epoch then
16  |   |   |   add randomly new weights (connections) in the same amount as the ones removed previously;
17  |   |   end
18  |   end
19  end
```

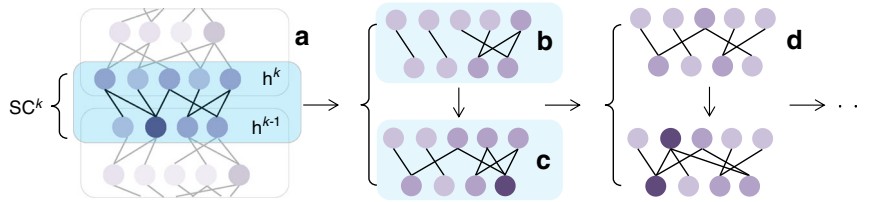

**Fig. 1** An illustration of the SET procedure. For each sparse connected layer, SC$^k$ (**a**), of an ANN at the end of a training epoch a fraction of the weights, the ones closest to zero, are removed (**b**). Then, new weighs are added randomly in the same amount as the ones previously removed (**c**). Further on, a new training epoch is performed (**d**), and the procedure to remove and add weights is repeated. The process continues for a finite number of training epochs, as usual in the ANNs training

**Table 1 Datasets characteristics**

| Experiments type | Dataset | | Dataset properties | | | | |
|---|---|---|---|---|---|---|---|
| | | | Domain | Data type | Features [#] | Train samples [#] | Test samples [#] |
| RBMs variants | UCI evaluation suite[65] | ADULT | Households | Binary | 123 | 5000 | 26,147 |
| | | Connect4 | Games | Binary | 126 | 16,000 | 47,557 |
| | | DNA | Genetics | Binary | 180 | 1400 | 1186 |
| | | Mushrooms | Biology | Binary | 112 | 2000 | 5624 |
| | | NIPS-0-12 | Documents | Binary | 500 | 400 | 1240 |
| | | OCR-letters | Letters | Binary | 128 | 32,152 | 10,000 |
| | | RCV1 | Documents | Binary | 150 | 40,000 | 150,000 |
| | | Web | Internet | Binary | 300 | 14,000 | 32,561 |
| | CalTech 101 Silhouettes[66] | 16 × 16 | Images | Binary | 256 | 4082 | 2302 |
| | | 28 × 28 | Images | Binary | 784 | 4100 | 2307 |
| | MNIST[67] | | Digits | Binary | 784 | 60,000 | 10,000 |
| MLPs variants | MNIST[67] | | Digits | Grayscale | 784 | 60,000 | 10,000 |
| | CIFAR10[68] | | Images | RGB colors | 3072 | 50,000 | 10,000 |
| | HIGGS[1] | | Particle physics | Real values | 28 | 10,500,000 | 500,000 |
| | Fashion-MNIST[69] | | Fashion products | Grayscale | 784 | 60,000 | 10,000 |
| CNNs variants | CIFAR10[68] | | Images | RGB colors | 3072 | 50,000 | 10,000 |

The data used in this paper have been chosen to cover a wide range of fields where ANNs have the potential to advance state-of-the-art, including biology, physics, computer vision, data mining, and economics

layers, trained with SET, can replace any fully-connected layers in ANNs, at no decrease in accuracy, while having quadratically fewer parameters even in the ANN design phase (before training). This leads to reduced memory requirements and may lead to quadratically faster computational times in both phases (i.e. training and inference). We demonstrate our claims on three popular ANN types (RBMs, multi-layer perceptrons (MLPs), and convolutional neural networks (CNNs)), on two types of tasks (supervised and unsupervised learning), and on 15 benchmark datasets. We hope that our approach will enable ANNs having billions of neurons and evolved topologies to be capable of handling complex real-world tasks that are intractable using state-of-the-art methods.

## Results

**SET method**. With SET, the bipartite ANN layers start from a random sparse topology (i.e. Erdős–Rényi random graph[24]), evolving through a random process during the training phase towards a scale-free topology. Remarkably, this process does not have to incorporate any constraints to force the scale-free topology. But our evolutionary algorithm is not arbitrary: it follows a phenomenon that takes place in real-world complex networks (such as biological neural networks and protein interaction networks). Starting from an Erdős–Rényi random graph topology and throughout millenia of natural evolution, networks end up

with a more structured connectivity, i.e. scale-free[7] or small-world[8] topologies.

The SET algorithm is detailed in Box 1 and exemplified in Fig. 1. Formally, let us define a sparse connected (SC$^k$) layer in an ANN. This layer has $n^k$ neurons, collected in a vector $\mathbf{h}^k = \left[ h_1^k, h_2^k, \ldots, h_{n^k}^k \right]$. Any neuron from $\mathbf{h}^k$ is connected to an arbitrary number of neurons belonging to the layer below, $\mathbf{h}^{k-1}$. The connections between the two layers are collected in a sparse weight matrix $\mathbf{W}^k \in \mathbf{R}^{n^{k-1} \times n^k}$. Initially, $\mathbf{W}^k$ is a Erdős–Rényi random graph, in which the probability of a connection between the neurons $h_i^k$ and $h_j^{k-1}$ is given by

$$p\left(W_{ij}^k\right) = \frac{\varepsilon\left(n^k + n^{k-1}\right)}{n^k n^{k-1}} \tag{1}$$

whereby $\varepsilon \in \mathbf{R}^+$ is a parameter of SET controlling the sparsity level. If $\varepsilon \ll n^k$ and $\varepsilon \ll n^{k+1}$ then there is a linear number of connections (i.e. non-zero elements), $n^W = |\mathbf{W}^k| = \varepsilon(n^k + n^{k-1})$, with respect to the number of neurons in the sparse layers. In the case of fully-connected layers the number of connections is quadratic, i.e. $n^k n^{k-1}$.

However, it may be that this random generated topology is not suited to the particularities of the data that the ANN model tries to learn. To overcome this situation, during the training process, after each training epoch, a fraction $\zeta$ of the smallest positive weights and of the largest negative weights of SC$^k$ is removed. These removed weights are the ones closest to zero, thus we do

not expect that their removal will notably change the model performance. This has been shown, for instance, in refs. [13,25] using more complex approaches to remove unimportant weights. Next, to let the topology of $SC^k$ to evolve so as to fit the data, an amount of new random connections, equal to the amount of weights removed previously, is added to $SC^k$. In this way, the number of connections in $SC^k$ remains constant during the training process. After the training ends, we keep the topology of $SC^k$ as the one obtained after the last weight removal step, without adding new random connections. To illustrate better these processes, we make the following analogy. If we assume a connection as the entity which evolves over time, the removal of the least important connections corresponds, roughly, to the selection phase of natural evolution, while the random addition of new connections corresponds, roughly, to the mutation phase of natural evolution.

It is worth highlighting that in the initial phase of conceiving the SET procedure, the weight-removal and weight-addition steps after each training epoch were introduced based on our own intuition. However, in the last phases of preparing this paper, we have found that there is a similarity between SET and a phenomenon which takes place in biological brains, named synaptic shrinking during sleep. This phenomenon has been demonstrated in two recent papers[26,27]. In short, it was found that during sleep the weakest synapses in the brain shrink, while the strongest synapses remain unaltered, supporting the hypothesis that one of the core functions of sleeping is to renormalize the overall synaptic strength increased while awake[27]. By keeping the

analogy, this is—in a way—what happens also with the ANNs during the SET procedure.

We evaluate SET in three types of ANNs, RBMs[28], MLPs, and CNNs[3] (all three are detailed in the Methods section), to experiment with both unsupervised and supervised learning. In total, we evaluate SET on 15 benchmark datasets, as detailed in Table 1, covering a wide range of fields in which ANNs are employed, such as biology, physics, computer vision, data mining, and economics. We also assess SET in combination with two different training methods, i.e. contrastive divergence[29] and SGD[3].

**Performance on RBMs**. First, we have analyzed the performance of SET on a bipartite undirected stochastic ANN model, i.e. RBM[28], which is popular for its unsupervised learning capability[30] and high performance as a feature extractor and density estimator[31]. The new model derived from the SET procedure was dubbed SET-RBM. In all experiments, we set $\varepsilon = 11$, and $\zeta = 0.3$, performing a small random search just on the MNIST dataset, to be able to assess if these two meta-parameters are dataset specific or if their values are general enough to perform well also on different datasets.

There are few studies on RBM connectivity sparsity[10]. Still, to get a good estimation of SET-RBM capabilities we compared it against RBM$_{\text{FixProb}}$[10] (a sparse RBM model with a fixed Erdős–Rényi topology), fully-connected RBMs, and with the state-of-the-art results of XBMs from ref. [10]. We chose

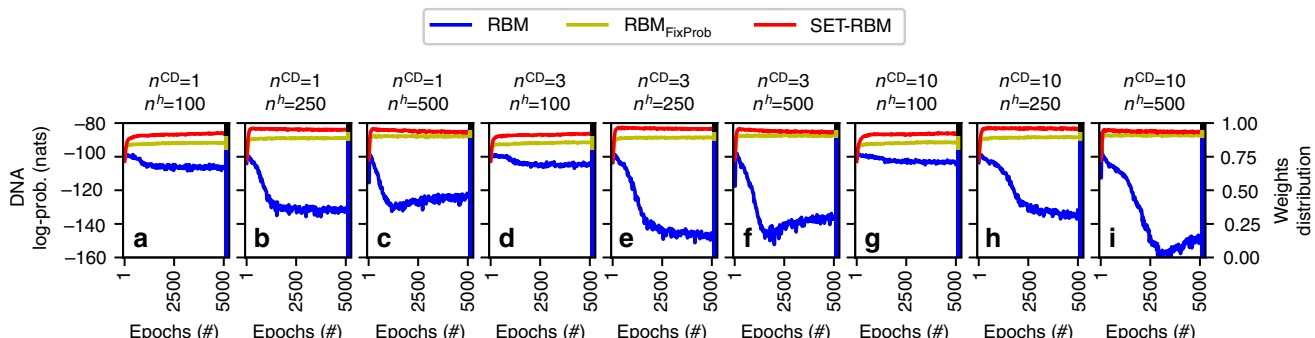

**Fig. 2** Experiments with RBM variants on the DNA dataset. For each model studied we have considered three cases for the number of contrastive divergence steps, $n^{CD} = 1$ (**a–c**), $n^{CD} = 3$ (**d–f**), and $n^{CD} = 10$ (**g–i**). Also, we considered three cases for the number of hidden neurons, $n^h = 100$ (**a, d, g**), $n^h = 250$ (**b, e, h**), and $n^h = 500$ (**c, f, i**). In each panel, the x axes show the training epochs; the left y axes show the average log-probabilities computed on the test data with AIS[33]; and the right y axes (the stacked bar on the right side of the panels) reflect the fraction given by the $n^W$ of each model over the sum of the $n^W$ of all three models. Overall, SET-RBM outperforms the other two models in most of the cases. Also, it is interesting to see that SET-RBM and RBM$_{\text{FixProb}}$ are much more stable and do not present the over-fitting problems of RBM

### Table 2 Summarization of the experiments with RBM variants

| Dataset | | RBM | | | | RBM$_{\text{FixProb}}$ | | | | SET-RBM | | | | XBM | | | |
|---|---|---|---|---|---|---|---|---|---|---|---|---|---|---|---|---|---|
| | | Log-prob. | $n^h$ | $n^W$ | $n^{CD}$ | Log-prob. | $n^h$ | $n^W$ | $n^{CD}$ | Log-prob. | $n^h$ | $n^W$ | $n^{CD}$ | Log-prob. | $n^h$ | $n^W$ | $n^{CD}$ |
| UCI evaluation suite | ADULT | −14.91 | 100 | 12,300 | 10 | −14.79 | 500 | 4984 | 10 | −13.85 | 500 | 4797 | 3 | −15.89 | 1200 | 12,911 | 1 |
| | Connect4 | −5.01 | 500 | 63,000 | 10 | −15.01 | 500 | 5008 | 10 | −13.12 | 500 | 4820 | 10 | −17.37 | 1200 | 12,481 | 1 |
| | DNA | −85.97 | 500 | 90,000 | 10 | −86.90 | 500 | 5440 | 10 | −82.51 | 250 | 3311 | 3 | −83.17 | 1600 | 17,801 | 1 |
| | Mushrooms | −11.35 | 100 | 11,200 | 10 | −11.36 | 500 | 4896 | 10 | −10.63 | 250 | 2787 | 10 | −14.71 | 1000 | 10,830 | 1 |
| | NIPS-0-12 | −274.60 | 250 | 125,000 | 3 | −282.67 | 500 | 8000 | 10 | −276.62 | 500 | 7700 | 3 | −287.43 | 100 | 5144 | 1 |
| | OCR-letters | −29.33 | 500 | 64,000 | 10 | −38.58 | 500 | 5024 | 10 | −28.69 | 500 | 4835 | 10 | −33.08 | 1200 | 13,053 | 1 |
| | RCV1 | −47.24 | 500 | 75,000 | 3 | −50.34 | 500 | 5200 | 10 | −47.60 | 500 | 5005 | 10 | −49.68 | 1400 | 14,797 | 1 |
| | Web | −31.74 | 500 | 150,000 | 1 | −31.32 | 500 | 6400 | 10 | −28.74 | 500 | 6160 | 10 | −30.62 | 2600 | 29,893 | 1 |
| CalTech 101 Silhouettes | 16 × 16 | −28.41 | 2500 | 640,000 | 10 | −53.25 | 5000 | 42,048 | 10 | −46.08 | 5000 | 40,741 | 10 | −69.29 | 500 | 6721 | 1 |
| | 28 × 28 | −159.51 | 5000 | 3,920,000 | 3 | −126.69 | 5000 | 46,272 | 10 | −104.89 | 2500 | 25,286 | 10 | −142.96 | 1500 | 19,201 | 1 |
| MNIST | | −91.70 | 2500 | 1,960,000 | 10 | −117.55 | 5000 | 46,272 | 10 | −86.41 | 5000 | 44,536 | 10 | −85.21 | 27,000 | 387,955 | 1:25 |

On each dataset, we report the best average log-probabilities obtained with AIS on the test data for each model. $n^h$ represents the number of hidden neurons, $n^{CD}$ the number of CD steps, and $n^W$ the number of weights in the model

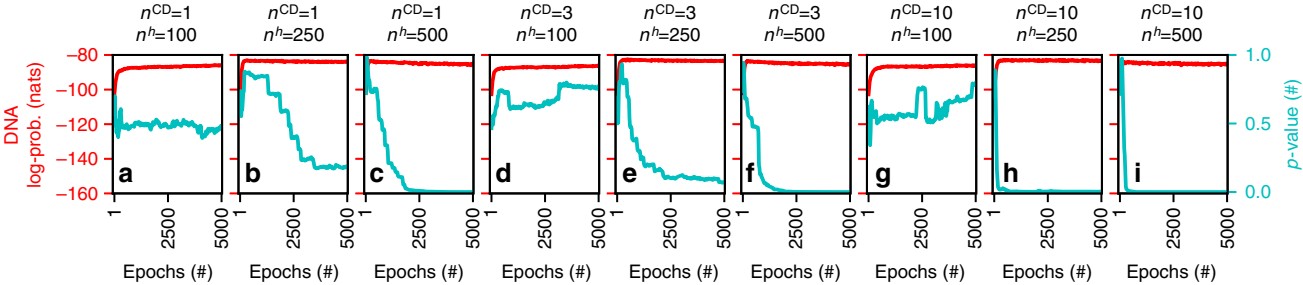

**Fig. 3** SET-RBM evolution towards a scale-free topology on the DNA dataset. We have considered three cases for the number of contrastive divergence steps, $n^{CD} = 1$ (**a–c**), $n^{CD} = 3$ (**d–f**), and $n^{CD} = 10$ (**g–i**). Also, we considered three cases for the number of hidden neurons, $n^h = 100$ (**a, d, g**), $n^h = 250$ (**b, e, h**), and $n^h = 500$ (**c, f, i**). In each panel, the x axes show the training epochs; the left y axes (red color) show the average log-probabilities computed for SET-RBMs on the test data with AIS[33]; and the right y axes (cyan color) show the p-values computed between the degree distribution of the hidden neurons in SET-RBM and a power-law distribution. We may observe that for models with a high enough number of hidden neurons, the SET-RBM topology always tends to become scale-free

RBM$_{FixProb}$ as a sparse baseline model to be able to understand better the effect of SET-RBM adaptive connectivity on its learning capabilities, as both models, i.e. SET-RBM and RBM$_{FixProb}$, are initialized with an Erdös–Rényi topology. We performed experiments on 11 benchmark datasets coming from various domains, as depicted in Table 1, using the same splitting for training and testing data as in ref. [10]. All models were trained for 5000 epochs using contrastive divergence[29] (CD) with 1, 3, and 10 CD steps, a learning rate of 0.01, a momentum of 0.9, and a weight decay of 0.0002, as discussed in ref. [32]. We evaluated the generative performance of the scrutinized models by computing the log-probabilities on the test data using annealed importance sampling (AIS)[33], setting all parameters as in refs. [10,33]. We have used MATLAB for this set of experiments. We implemented SET-RBM and RBM$_{FixProb}$ ourselves; while for RBM and AIS we have adapted the code provided by Salakhutdinov and Murray[33].

Figure 2 depicts the model's performance on the DNA dataset; while Supplementary Fig. 1 presents results on all datasets, using varying numbers of hidden neurons (i.e. 100, 250, and 500 hidden neurons for the UCI evaluation suite datasets; and 500, 2500, and 5000 hidden neurons for the CalTech 101 Silhouettes and MNIST datasets). Table 2 summarizes the results, presenting the best performer for each type of model for each dataset. In 7 out of 11 datasets, SET-RBM outperforms the fully-connected RBM, while reducing the parameters by a few orders of magnitude. For instance, on the MNIST dataset, SET-RBM reaches −86.41 nats (natural units of information), with a 5.29-fold improvement over the fully-connected RBM, and a parameters reduction down to 2%. In 10 out of 11 datasets, SET-RBM outperforms XBM, which represents the state-of-the-art results on these datasets for sparse variants of RBM[10]. It is interesting to see in Table 2 that RBM$_{FixProb}$ reaches its best performance on each dataset in the case when the maximum number of hidden neurons is considered. Even if SET-RBM has the same amount of weights with RBM$_{FixProb}$, it reaches its maximum performance on 3 out of the 11 datasets studied just when a medium number of hidden neurons is considered (i.e. DNA, Mushrooms, and CalTech 101 Silhouettes 28 × 28).

Figure 2 and Supplementary Fig. 1 present striking results on stability. Fully-connected RBMs show instability and over-fitting issues. For instance, using one CD step on the DNA dataset the RBMs have a fast learning curve, reaching a maximum after several epochs. After that, the performance start to decrease giving a sign that the models start to be over-fitted. Moreover, as expected, the RBM models with more hidden neurons (Fig. 2b, c, e, f, h, i) over-fit even faster than the one with less hidden neurons (Fig. 2a, d, g). A similar behavior can be seen in most of the cases

considered, culminating with a very spiky learning behavior in some of them (Supplementary Fig. 1). Contrary to fully-connected RBMs, the SET procedure stabilizes SET-RBMs and avoids over-fitting. This situation can be observed more often when a high number of hidden neurons is chosen. For instance, if we look at the DNA dataset, independently on the values of $n^h$ and $n^{CD}$ (Fig. 2), we may observe that SET-RBMs are very stable after they reach around −85 nats, having almost a flat learning behavior after that point. Contrary, on the same dataset, the fully-connected RBMs have a very short initial good learning behavior (for few epochs) and, after that, they go up and down during the 5000 epochs analyzed, reaching the minimum performance of −160 nats (Fig. 2i). Note that these good stability and over-fitting avoidance capacities are induced not just by the SET procedure, but also by the sparsity itself, as RBM$_{FixProb}$, too, has a stable behavior in almost all the cases. This happens due to the very small number of optimized parameters of the sparse models in comparison with the high number of parameters of the fully-connected models (as reflected by the stacked bar from the right y-axis of each panel of Fig. 2 and Supplementary Fig. 1) which does not allow the learning procedure to over-fit the sparse models on the training data.

Furthermore, we verified our initial hypothesis about sparse connectivity in SET-RBM. Figure 3 and Supplementary Fig. 2 show how the hidden neurons' connectivity naturally evolves towards a scale-free topology. To assess this fact, we have used the null hypothesis from statistics[34], which assumes that there is no relation between two measured phenomena. To see if the null hypothesis between the degree distribution of the hidden neurons and a power-law distribution can be rejected, we have computed the p-value[35,36] between them using a one-tailed test. To reject the null hypothesis the p-value has to be lower than a statistically significant threshold of 0.05. In all cases (all panels of Fig. 3), looking at the p-values (y axes to the right of the panels), we can see that at the beginning of the learning phase the null hypothesis is not rejected. This was to be expected, as the initial degree distribution of the hidden neurons is binomial due to the randomness of the Erdös–Rényi random graphs[37] used to initialize the SET-RBMs topology. Subsequently, during the learning phase, we can see that, in many cases, the p-values decrease considerably under the 0.05 threshold. When these situations occur, it means that the degree distribution of the hidden neurons in SET-RBM starts to approximate a power-law distribution. As to be expected, the cases with fewer neurons (Fig. 3a, b, d, e, g) fail to evolve to scale-free topologies, while the cases with more neurons always evolve towards a scale-free topology (Fig. 3c, f, h, i). To summarize, in 70 out of 99 cases studied (all panels of Supplementary Fig. 2), the SET-RBMs

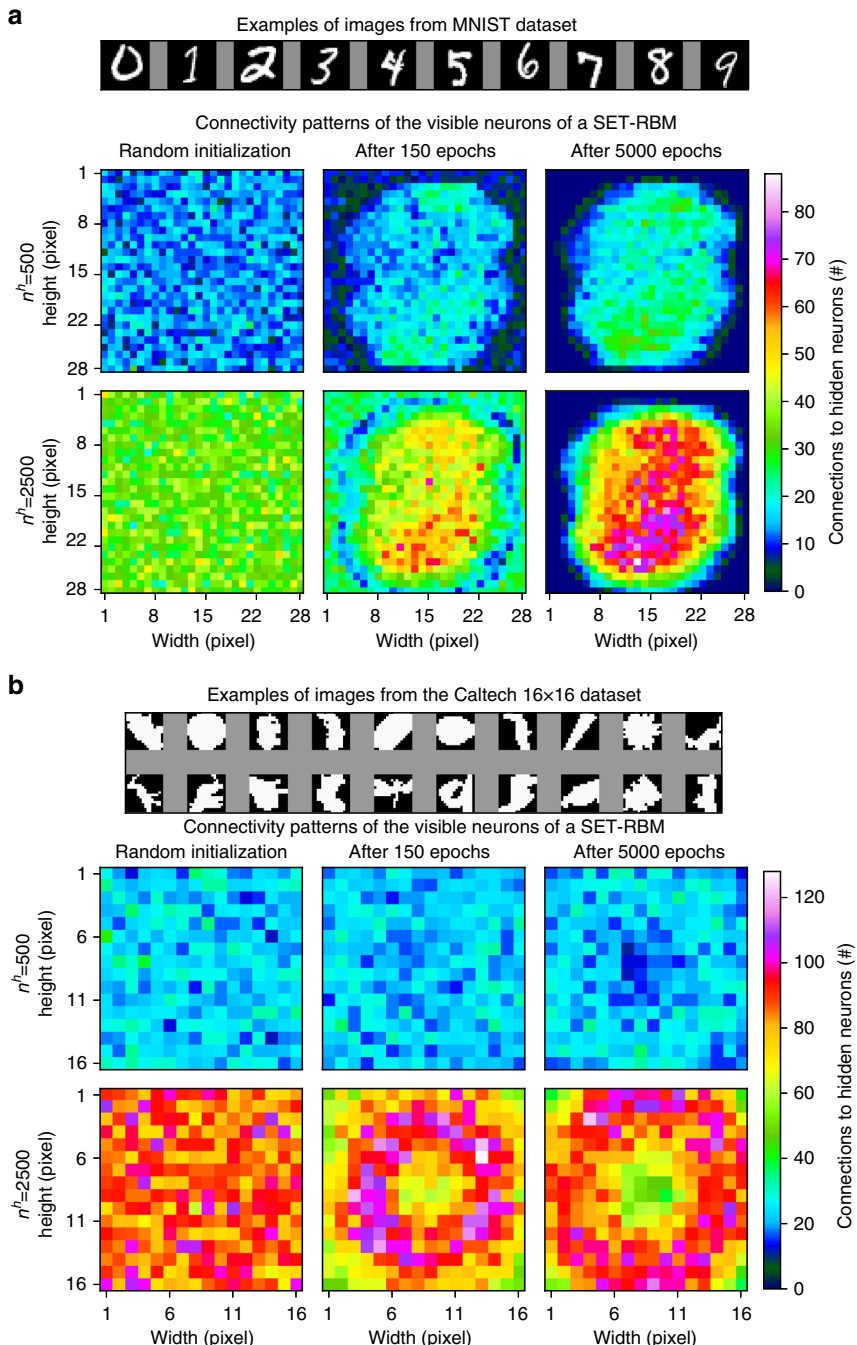

**Fig. 4** SET-RBMs connectivity patterns for the visible neurons. **a** On the MNIST dataset. **b** On the Caltech 101 16 × 16 dataset. For each dataset, we have analyzed two SET-RBM architectures, i.e. 500 and 2500 hidden neurons. The heat-map matrices are obtained by reshaping the visible neurons vector to match the size of the original input images. In all cases, it can be observed that the connectivity starts from an initial Erdös–Rényi distribution. Then, during the training process, it evolves towards organized patterns which depend on the input images

hidden neurons' connectivity evolves clearly during the learning phase from an Erdös–Rényi topology towards a scale-free one.

Moreover, in the case of the visible neurons, we have observed that their connectivity tends to evolve into a pattern that is dependent on the domain data. To illustrate this behavior, Fig. 4 shows what happens with the amount of connections for each visible neuron during the SET-RBM training process on the MNIST and CalTech 101 datasets. It can be observed that initially the connectivity patterns are completely random, as given by the binomial distribution of the Erdös–Rényi topology. After the

models are trained for several epochs, some visible neurons start to have more connections and others fewer and fewer. Eventually, at the end of the training process, some clusters of the visible neurons with clearly different connectivities emerge. Looking at the MNIST dataset, we can observe clearly that in both cases analyzed (i.e. 500 and 2500 hidden neurons) a cluster with many connections appeared in the center. At the same time, on the edges, another cluster appeared in which each visible neuron has zero or very few connections. The cluster with many connections corresponds exactly to the region where the digits appear in the images. On the Caltech 101 dataset, a similar behavior can be

**Table 3 Summarization of the experiments with MLP variants**

| Dataset | Data augmentation | Architecture | Activation | MLP | | MLP_{FixProb} | | SET-MLP | |
|---|---|---|---|---|---|---|---|---|---|
| | | | | Accuracy [%] | $n^W$ | Accuracy [%] | $n^W$ | Accuracy [%] | $n^W$ |
| MNIST | No | 784-1000-1000-1000-10 | SReLU | 98.55 | 2,794,000 | 97.68 | 89,797 | 98.74 | 89,797 |
| CIFAR10 | Yes | 3072-4000-1000-4000-10 | SReLU | 68.70 | 20,328,000 | 62.19 | 278,630 | 74.84 | 278,630 |
| HIGGS | No | 28-1000-1000-1000-2 | SReLU | 78.44 | 2,038,000 | 76.69 | 80,614 | 78.47 | 80,614 |

On each dataset, we report the best classification accuracy obtained by each model on the test data. $n^W$ represents the number of weights in the model. The only difference between the three models is the network topology, i.e. MLP has fully connected layers, MLP_{FixProb} has sparse layers with Erdös–Rényi fixed topology, and SET-MLP has sparse evolutionary layers trained with SET

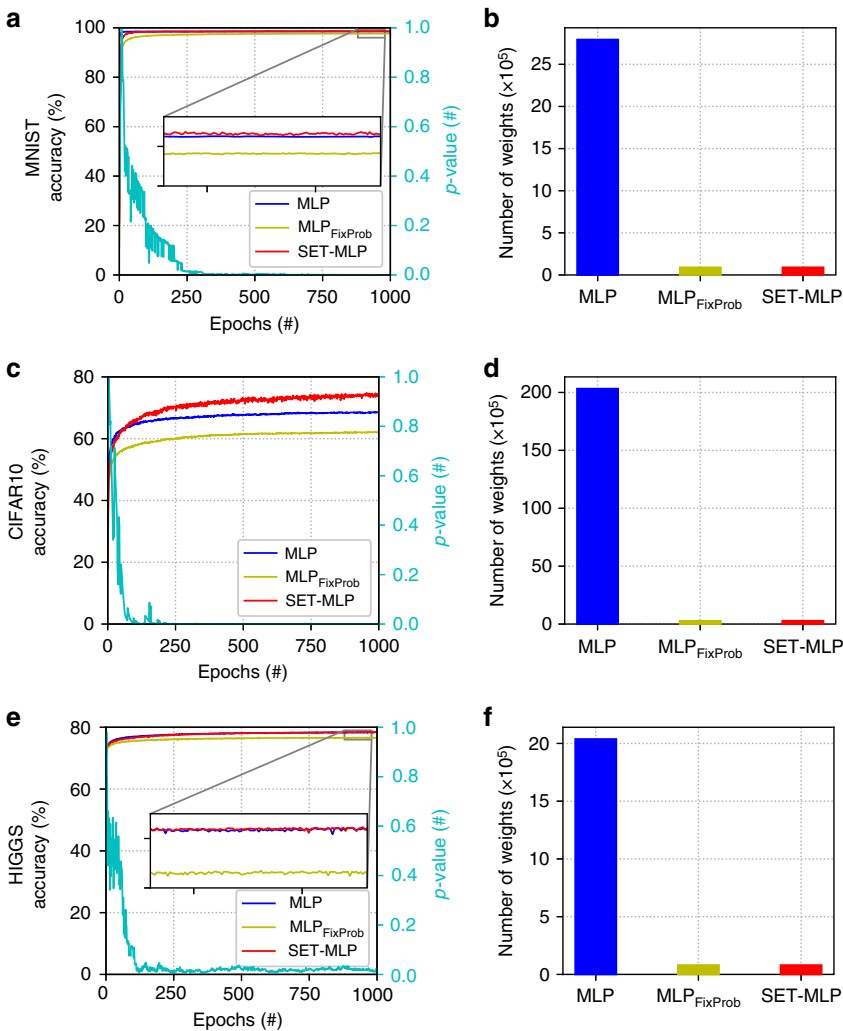

**Fig. 5** Experiments with MLP variants using three benchmark datasets. **a**, **c**, **e** reflect models performance in terms of classification accuracy (left *y* axes) over training epochs (*x* axes); the right *y* axes of **a**, **c**, **e** give the *p*-values computed between the degree distribution of the hidden neurons of the SET-MLP models and a power-law distribution, showing how the SET-MLP topology becomes scale-free over training epochs. **b**, **d**, **f** depict the number of weights of the three models on each dataset. The most striking situation happens for the CIFAR10 dataset (**c**, **d**) where the SET-MLP model outperforms drastically the MLP model, while having ~100 times fewer parameters

observed, except the fact that due to the high variability of shapes on this dataset the less connected cluster still has a considerable amount of connections. This behavior of the visible neurons' connectivity may be used, for instance, to perform dimensionality reduction by detecting the most important features on high-dimensional datasets, or to make faster the SET-RBM training process.

**Performance on MLPs**. To better explore the capabilities of SET, we have also assessed its performance on classification tasks based on supervised learning. We developed a variant of MLP[3], dubbed

SET-MLP, in which the fully-connected layers have been replaced with sparse layers obtained through the SET procedure, with $\varepsilon = 20$, and $\zeta = 0.3$. We kept the $\zeta$ parameter as in the previous case of SET-RBM, while for the $\varepsilon$ parameter we performed a small random search just on the MNIST dataset. We compared SET-MLP to a standard fully-connected MLP, and to a sparse variant of MLP having a fixed Erdős–Rényi topology, dubbed MLP_{FixProb}. For the assessment, we have used three benchmark datasets (Table 1), two coming from the computer vision domain (MNIST and CIFAR10), and one from particle physics (the HIGGS dataset[1]). In all cases, we have used the same data processing

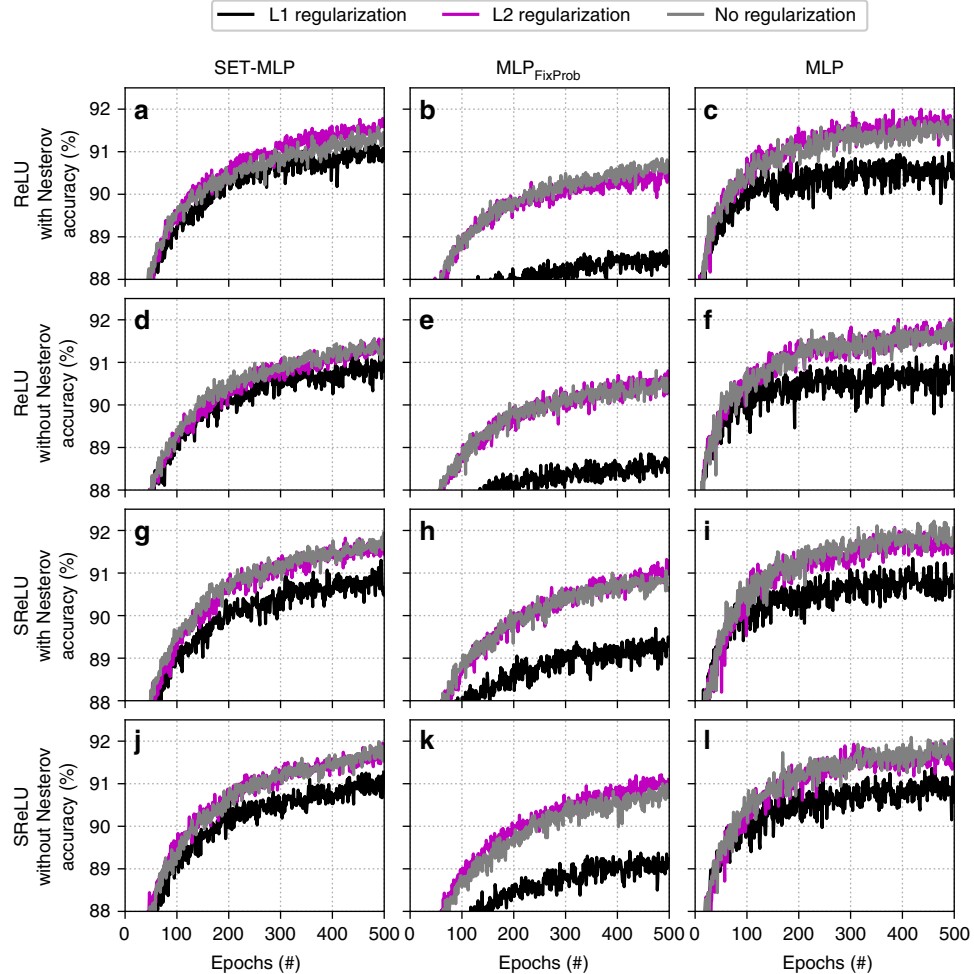

**Fig. 6** Models accuracy using three weights regularization techniques on the Fashion-MNIST dataset. All models have been trained with stochastic gradient descent, having the same hyper-parameters, number of hidden layers (i.e. three), and number of hidden neurons per layer (i.e. 1000). **a–c** use ReLU activation function for the hidden neurons and Nesterov momentum; **d–f** use ReLU activation function without Nesterov momentum; **g–i** use SReLU activation function and Nesterov momentum; and **j–l** use SReLU activation function without Nesterov momentum. **a**, **d**, **g**, **j** present experiments with SET-MLP; **b**, **e**, **h**, **k** with MLP$_{FixProb}$; and **c**, **f**, **i**, **l** with MLP

techniques, network architecture, training method (i.e. SGD[3] with fixed learning rate of 0.01, momentum of 0.9, and weight decay of 0.0002), and a dropout rate of 0.3 (Table 3). The only difference between MLP, MLP$_{FixProb}$, and SET-MLP, consisted in their topological connectivity. We have used Python and the Keras library (https://github.com/fchollet/keras) with Theano backend[38] for this set of experiments. For MLP we have used the standard Keras implementation, while we implemented ourselves SET-MLP and MLP$_{FixProb}$ on top of the standard Keras libraries.

The results depicted in Fig. 5 show how SET-MLP outperforms MLP$_{FixProb}$. Moreover, SET-MLP always outperforms MLP, while having two orders of magnitude fewer parameters. Looking at the CIFAR10 dataset, we can see that with only just 1% of the weights of MLP, SET-MLP leads to significant gains. At the same time, SET-MLP has comparable results with state-of-the-art MLP models after these have been carefully fine tuned. To quantify, the second best MLP model in the literature on CIFAR10 reaches about 74.1% classification accuracy[39] and has 31 million parameters: while SET-MLP reaches a better accuracy (74.84%) having just about 0.3 million parameters. Moreover, the best MLP model in the literature on CIFAR10 has 78.62% accuracy[40], with about 12 million parameters, while also benefiting from a pre-training phase[41,42]. Although we have not pre-trained the MLP

models studied here, we should mention that SET-RBM can be easily used to pre-train a SET-MLP model to further improve performance.

With respect to the stability and over-fitting issues, Fig. 5 shows that SET-MLP is also very stable, similarly to SET-RBM. Note that due to the use of the dropout technique, the fully-connected MLP is also quite stable. Regarding the topological features, we can see from Fig. 5 that, similarly to what was found in the SET-RBM experiments (Fig. 3), the hidden neuron connections in SET-MLP rapidly evolve towards a power-law distribution.

To understand better the effect of various regularization techniques, and activation functions, we performed a small controlled experiment on the Fashion-MNIST dataset. We chose this dataset because it has a similar size with the MNIST dataset, being at the same time a harder classification problem. We used MLP, MLP$_{FixProb}$, and SET-MLP with three hidden layers of 1000 hidden neurons each. Then, we varied for each model the following: (1) the weights regularization method (i.e. L1 regularization with a rate of 0.0000001, L2 regularization with a rate of 0.0002, and no regularization), (2) the use (or not use) of Nesterov momentum, and (3) two activation functions (i.e. SReLU[43] and ReLU[44]). The regularization rates were found by

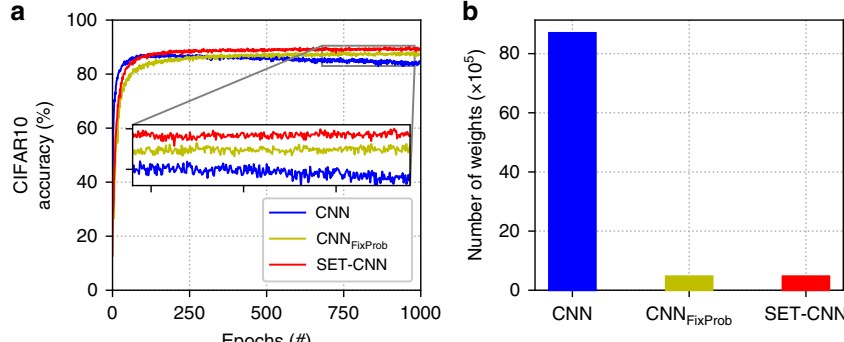

**Fig. 7** Experiments with CNN variants on the CIFAR10 dataset. **a** Models performance in terms of classification accuracy (left y axes) over training epochs (x axes). **b** The number of weights of the three models on each dataset. The convolutional layers of each model have in total 287,008 weights, while the fully connected (or the sparse) layers on top have 8,413,194, 184.842, and 184,842 weights for CNN, CNN$_{FixProb}$, and SET-CNN, respectively

performing a small random search procedure with L1 and L2 levels between 0.01 and 0.0000001 to try maximizing the performance of all the three models. In all cases, we used SGD with 0.01 learning rate to train the models. The results depicted in Fig. 6 show that, in this specific scenario, SET-MLP achieves the best performance if no regularization or L2 regularization is used for the weights, while L1 regularization does not offer the same level of performance. To summarize, SET-MLP achieves the best results on the Fashion-MNIST dataset with the following settings: SReLU activation function, without Nesterov momentum, and without (or with L2) weights regularization. These being, in fact, the settings that we used in the MLP experiments discussed above. It is worth highlighting that independently on the specific setting, the general conclusion drawn up to now still holds. SET-MLP achieves a similar (or better) performance to that of MLP, while having a much smaller number of connections. Also, SET-MLP always clearly outperforms MLP$_{FixProb}$.

**Performance on CNNs**. As one of the most used ANN models nowadays are CNNs[3], we have briefly studied how SET can be used in the CNN architectures to replace their fully connected layers with sparse evolutionary counterparts. We considered a standard small CNN architecture, i.e. conv(32,(3,3))-dropout (0.3)-conv(32,(3,3))-pooling-conv(64,(3,3))-dropout(0.3)-conv (64,(3,3))-pooling-conv(128,(3,3))-dropout(0.3)-conv(128,(3,3))-pooling), where the numbers in brackets for the convolutional layers mean (number of filters, (kernel size)), and for the dropout layers represent the dropout rate. Then, on top of the convolutional layers, we have used: (1) two fully connected layers to create a standard CNN, (2) two sparse layers with a fixed Erdős–Rényi topology to create a CNN$_{FixProb}$, and (3) two evolutionary sparse layers to create a SET-CNN. For each model, each of the two layers on top was followed by a dropout (0.3) layer. On top of these, the CNN, CNN$_{FixProb}$, and SET-CNN contained also a softmax layer. Even if SReLU seems to offer a slightly better performance, we used ReLU as activation function for the hidden neurons due to its wide utilization. We used SGD to train the models. The experiments were performed on the CIFAR10 dataset. The results are depicted in Fig. 7. They show, same as in the previous experiments with restricted Boltzmann machine and multi-layer perceptron, that SET-CNN can achieve a better accuracy than CNN, even if it has just about 4% of the CNN connections. To quantify this, we mention that in our experiments SET-CNN reaches a maximum of 90.02% accuracy, CNN$_{FixProb}$ achieves a maximum of 88.26% accuracy, while CNN achieves a maximum of 87.48% accuracy. Similar with the RBM experiments, we can observe that CNN is subject to a small over-

fitting behavior, while CNN$_{FixProb}$ and SET-CNN are very stable. Even if our goal was just to show that SET can be combined also with the widely used CNNs and not to optimize the CNN variants architectures to increase the performance, we highlight that, in fact, SET-CNN achieves a performance comparable with state-of-the-art results. The benefit of using SET in CNNs is two-fold: to reduce the total number of parameters in CNNs and to permit the use of larger CNN models.

Last but not least, during all the experiments performed, we observed that SET is quite stable with respect to the choice of meta-parameters $\varepsilon$ and $\zeta$. There is no way to say that our choices offered the best possible performance, even if we fine-tuned them just on one dataset, i.e. MNIST, and we evaluated their performance on all 15 datasets. Still, we can say that a $\zeta = 0.3$ for both, SET-RBM and SET-MLP, and an $\varepsilon$ specific for each model type, SET-RBM ($\varepsilon = 11$), SET-MLP ($\varepsilon = 20$), and SET-CNN ($\varepsilon = 20$) were good enough to outperform state-of-the-art.

Considering the different datasets under scrutiny, we stress that we have assessed both image-intensive and non-image sets. On image datasets, CNNs[3] typically outperform MLPs. However, CNNs are not viable on other types of high-dimensional data, such as biological data (e.g.[45]), or theoretical physics data (e.g.[1]). In those cases, MLPs will be a better choice. This is in fact the case of the HIGGS dataset (Fig. 5e, f), where SET-MLP achieves 78.47% classification accuracy and has about 90,000 parameters. Whereas, one of the best MLP models in the literature achieved a 78.54% accuracy, while having three times more parameters[40].

## Discussion

In this paper, we have introduced SET, a simple and efficient procedure to replace ANNs' fully-connected bipartite layers with sparse layers. We have validated our approach on 15 datasets (from different domains) and on three widely used ANN models, i.e. RBMs, MLPs, and CNNs. We have evaluated SET in combination with two different training methods, i.e. contrastive divergence and SGD, for unsupervised and supervised learning. We showed that SET is capable of quadratically reducing the number of parameters of bipartite neural networks layers from the ANN design phase, at no decrease in accuracy. In most of the cases, SET-RBMs, SET-MLPs, and SET-CNNs outperform their fully-connected counterparts. Moreover, they always outperform their non-evolutionary counterparts, i.e. RBM$_{FixProb}$, MLP$_{FixProb}$, and CNN$_{FixProb}$.

We can conclude that the SET procedure is coherent with real-world complex networks, whereby nodes' connections tend to evolve into scale-free topologies[46]. This feature has important

implications in ANNs: we could envision a computational time reduction by reducing the number of training epochs, if we would use for instance preferential attachment algorithms[47] to evolve faster the topology of the bipartite ANN layers towards a scale-free one. Of course, this possible improvement has to be treated carefully, as forcing the model topology to evolve unnaturally faster into a scale-free topology may be prone to errors—for instance, the data distribution may not be perfectly matched. Another possible improvement would be to analyze how to remove the unimportant weights. In this article, we showed that it is efficient for SET to directly remove the connections with weights closest to zero. Note that we have tried also to remove connections randomly, and, as expected, this led to dramatic reductions in accuracy. Likewise, when we tried to remove the connections with the largest weights, the SET-MLP model was not able to learn at all, performing similarly to a random classifier. However, we do not exclude the possibility that there may be better, more sophisticated approaches to removing connections, e.g. using gradient methods[25], or centrality metrics from network science[48].

SET can be widely adopted to reduce the fully-connected layers into sparse topologies in other types of ANNs, e.g., recurrent neural networks[3], deep reinforcement learning networks[2,49], and so on. For a large scale utilization of SET, from the academic environment to industry, one more step has to be achieved. Currently, all state-of-the-art deep learning implementations are based on very well-optimized dense matrix multiplications on graphics processing units (GPUs), while sparse matrix multiplications are extremely limited in performance[50,51]. Thus, until optimized hardware for SET-like operations will appear (e.g., sparse matrix multiplications), one would have to find some alternative solutions. E.g., low-level parallel computations of neurons activations based just on their incoming connections and data batches to still perform dense matrix multiplications and to have a low-memory footprint. If these software engineering challenges are solved, SET may prove to be the basis for much larger ANNs, perhaps on a billion-node scale, to run in super-computers. Also, it may lead to the building of small but powerful ANNs, which could be directly trained on low-resource devices (e.g. wireless sensor nodes, mobile phones), without the need of first training them on supercomputers and then to move the trained models to low-resource devices, as is currently done by state-of-the-art approaches[13]. These powerful capabilities will be enabled by the linear relation between the number of neurons and the amount of connections between them yielded by SET. ANNs built with SET will have much more representational power, and better adaptive capabilities than the current state-of-the-art ANNs, and we hope that they will create a new research direction in artificial intelligence.

## Methods

**Artificial neural networks**. ANNs[52] are mathematical models, inspired by biological neural networks, which can be used in all three machine learning paradigms (i.e. supervised learning[53], unsupervised learning[53], and reinforcement learning[54]). These make them very versatile and powerful, as quantifiable by the remarkable success registered recently by the last generation of ANNs (also known as deep ANNs or deep learning[3]) in many fields from computer vision[3] to gaming[2,49]. Just like their biological counterparts, ANNs are composed by neurons and weighted connections between these neurons. Based on their purposes and architectures, there are many models of ANNs, such as RBMs[28], MLPs[55], CNNs[56], recurrent neural networks[57], and so on. Many of these ANN models contain fully-connected layers. A fully-connected layer of neurons means that all its neurons are connected to all the neurons belonging to its adjacent layer in the ANN architecture. For the purpose of this paper, in this section we briefly describe three models that contain fully-connected layers, i.e. RBMs[28], MLPs[55], and CNNs[3].

A restricted Boltzmann machine is a two-layer, generative, stochastic neural network that is capable to learn a probability distribution over a set of inputs[28] in an unsupervised manner. From a topological perspective, it allows only interlayer connections. Its two layers are: the visible layer, in which the neurons represent the

input data; and the hidden layer, in which the neurons represent the features automatically extracted by the RBM model from the input data. Each visible neuron is connected to all hidden neurons through a weighted undirected connection, leading to a fully-connected topology between the two layers. Thus, the flow of information is bidirectional in RBMs, from the visible layer to the hidden layer, and from the hidden layer to the visible layer, respectively. RBMs, beside being very successful in providing very good initialization weights to the supervised training of deep artificial neural network architectures[42], are also very successful as stand alone models in a variety of tasks, such as density estimation to model human choice[31], collaborative filtering[58], information retrieval[59], multi-class classification[60], and so on.

Multi-Layer Perceptron[55] is a classical feed-forward ANN model that maps a set of input data to the corresponding set of output data. Thus, it is used for supervised learning. It is composed by an input layer in which the neurons represent the input data, an output layer in which the neurons represent the output data, and an arbitrary number of hidden layers in between, with neurons representing the hidden features of the input data (to be automatically discovered). The flow of information in MLPs is unidirectional, starting from the input layer towards the output layer. Thus, the connections are unidirectional and exist just between consecutive layers. Any two consecutive layers in MLPs are fully-connected. There are no connections between the neurons belonging to the same layer, or between the neurons belonging to layers which are not consecutive. In ref. [61], it has been demonstrated that MLPs are universal function approximators, so they can be used to model any type of regression or classification problems.

CNNs[3] are a class of feed-forward neural networks specialized for image recognition, representing the state-of-the-art on these type of problems. They typically contain an input layer, an output layers, and a number of hidden layers in between. From bottom to top, the first hidden layers are the convolutional layers, inspired by the biological visual cortex, in which each neuron receives information just from the previous layer neurons belonging to its receptive field. Then, the last hidden layers are the fully connected ones.

In general, working with ANN models involves two phases: (1) training (or learning), in which the weighted connections between neurons are optimized using various algorithms (e.g. backpropagation procedure combined with SGD[62,63] used in MLPs or CNNs, contrastive divergence[29] used in RBMs) to minimize a loss function defined by their purpose; and (2) inference, in which the optimized ANN model is used to fulfill its purpose.

**Scale-free complex networks**. Complex networks (e.g. biological neural networks, actors and movies, power grids, transportation networks) are everywhere, in different forms, and different fields (from neurobiology to statistical physics[4]). Formally, a complex network is a graph with non-trivial topological features, human-made or nature-made. One of the most well-known and deeply studied type of topological features in complex networks is scale-freeness, due to the fact that a wide range of real-world complex networks have this topology. A network with a scale-free topology[7] is a sparse graph[64] that approximately has a power-law degree distribution $P(d) \sim d^{-\gamma}$, where the fraction $P(d)$ from the total nodes of the network has $d$ connections to other nodes, and the parameter $\gamma$ usually stays in the range $\gamma \in (2, 3)$.

**Data availability**. The data used in this paper are public datasets, freely available online, as reflected by their corresponding citations from Table 1. Prototype software implementations of the models used in this study are freely available online at https://github.com/dcmocanu/sparse-evolutionary-artificial-neural-networks

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

## Author contributions

D.C.M. and E.M. conceived the initial idea. D.C.M., E.M., P.S., P.H.N., M.G., and A.L. designed the experiments and analyzed the results. D.C.M. performed the experiments. D.C.M., E.M., P.S., P.H.N., M.G., and A.L. wrote the manuscript.

## Additional information

**Competing interests:** The authors declare no competing interests.

