## [Peer Review File · Nature Communications]

Reviewers' comments:

Reviewer #1 (Remarks to the Author):

Probably the most important idea behind this paper is that neural networks can be designed as sparse scale-free networks, and that this does not necessarily hurt what can be learned, that learning can be faster, and that this can have a positive impact on generalization. All of these things are important.

The paper has weaknesses however. The sparseness ideas advanced by the paper (by itself) are fine. But what the authors is doing is not "Evolutionary Training" and is at best a very crude kind of local search (local random sampling is probably a better description). In addition to sparseness, the authors are removing edges with weights that are close to zero, and also adding in new random weights. A more sophisticated version of this idea (albeit one-sided) is the "Weight Elimination" methods introduced by Wiegand/Rumelhart/Huberman (NIPS 1991), which used gradient methods to remove edges with weights near zero. There have also been more complex methods using genetic algorithms to do this sort of thing, but much of this work was also in the 1990s and at the time, was only applied to small problems. The problem with this early work was scalability, but this work is getting attention again and the scalability issues have changed (obviously). What is being done here is potentially important, but it is being done in a very simplistic manner.

Still, the sparseness idea is important, and it is getting more attention. (I have been working with very sparse neural networks for at least 3 years.)

The current paper is a sloppy (typos, poorly constructed sentences, jargon) and the experimental data is presented as a "data dump" that most readers will probably find hard to interpret. The authors would have been much better off presenting less data, and rather provide examples that better illustrate the effectiveness of the methods. Or results could have been presented in a more compact fashion.

Most of the paper is probably not readable to the average reader in its current form.

The quality of the presentation is less than what I would expect from a reasonable conference publication in a good venue (e.g. NIPS or AAAI or IJCNN).

From the abstract:

“fully connected”

I don't think there are any successful applications of reinforcement learning.

Well, at least not of the TD, Q-learning flavor.

page 1: “we have hinted a similar fact”

page 2: “highest negative weights” surely, not what you meant.

Page 2: the comments about the “selection phase of natural selection” are just not accurate.

On page 3, I am not sure why the authors are talking about performance in terms of “nats” instead of accuracy. (I asked around to see if anyone else had seen “nats” used in this way; no one had. Still, this may be ignorance on my part.)

Reviewer #2 (Remarks to the Author):

The paper presents an evolutionary-inspired training method to create sparse networks (for Boltzmann machines and MLPs), which are then in turn trained through unsupervised/supervised training methods.

The novel contribution of the paper is to show that ANNs can in fact perform well with sparsely-connected layers on a variety of different problems. This indicates that not as many ANN parameters are actually needed as are used in most current more fully-connected networks. Interestingly, the

approach could also be used to estimate the real problem dimensionality of a problem by counting the number of remaining weights.

To improve the manuscript I would suggest the following:

- The mentioned neuroevolution (NE) examples seem a little arbitrary. For example, the popular NEAT method is not mentioned. Additionally, most NE methods do evolve sparse networks but differently than the approach in this paper. It would be good to highlight the differences in this paper to the traditional use of evolution for neural networks (which involves adding nodes and connections over evolutionary time).
- It would be interesting to see example of the evolved weight patterns (maybe for MNIST). Are there any regularities in these patterns? Do they depend on the domain?
- Ideally, it would be good to see how the approach works together with the more commonly used convolutional networks (e.g. only replacing the last fully connected layers with SET) for image-based domains. Is there still an advantage in those cases? If this was the case it would further increase the impact of the paper.

Minor comments:

- Which test is used to determine the p-values? Does the test rely on normal distributions?
- The network notation in Table 3 is never really described. I assume all layers are fully-connected so each layer is replaced by SET?

In conclusion, an interesting paper that raises an important point about the current general practice in neural networks.

Reviewer #3 (Remarks to the Author):

As listed in the abstract, the major claims of the paper are (1) that network sparsity is unappreciated within neural network training, that (2) the proposed SET method reduces parameters and computation time quadratically relative to standard training approaches while improving performance, and that (3) the method will enable billion-neuron networks that are intractable with current methods.

The paper is interesting, the approach seems promising as viewed by the results, but the claims are reaching and generally undersupported relative to their extremity. As a result, I recommend the authors revise and resubmit their paper.

For the first claim: The value of sparsity is not unknown within deep learning. Drop-out of FC is common (and is used in the experiments), and in effect trains an ensemble of sparse networks to great effect (although the level of sparsity is less than that advocated in this paper). L1 regularization is well-known and induces pressure towards sparsity -- the authors should perform a control using L1 (the authors use weight decay, which is L2 regularization, and is not motivated directly by sparsity). For more sophisticated approaches to sparsity, see <https://arxiv.org/pdf/1702.06257.pdf> and <https://arxiv.org/pdf/1412.1442.pdf>.

For the second claim: It is interesting that the methods can greatly reduce the number of trainable parameters (although this potential is already established as the authors note in the network compression literature), in particular that you can start training sparsely instead of starting FC and going towards sparse. However, this prior precedent should be cited:

<https://arxiv.org/pdf/1606.06216.pdf>. More problematic is the claim that computation time can be similarly reduced -- this requires empirical support. GPUs are not optimized for sparse calculations, and it is not clear that the potential for computational acceleration can actually be realized (see <https://arxiv.org/pdf/1506.02515.pdf> and <https://arxiv.org/pdf/1702.06257.pdf> to see methods informed by such limitations). Timing information from Keras on GPUs should be provided if the method really does decrease computational costs quadratically as claimed.

For the third claim: The hard part of any method is scaling it -- raw extrapolation is nearly always deceiving. That this method really will enable billion-neuron networks where other methods would not seems like something that it would be wise to be skeptical of (given that the networks in the paper are on the order of ~10,000 nodes, several orders of magnitude shy of one billion). To claim that the method will enable solving currently-intractable tasks when it has not been applied on tasks of moderate complexity like imagenet also seems like a large stretch, where raw extrapolation is more likely than not deceiving. The ANN literature is littered with ideas that seemed promising at small scales but did not scale.

I can imagine readers will find this paper interesting, as I wouldn't have anticipated that the method would work as well as it does, and it is promising that it works both for RBMs and for MLPs, and achieves good results on permutation invariant MNIST and CIFAR-10. I'm curious to see how it scales and pairs with convolution, but I do not foresee it much influencing the field at large yet -- there are many papers targeting sparsity and compression, and usually the ML field is moved by big results (for better or worse) -- i.e. state of the art in a relevant domain, which this paper lacks.

Some nit-picks -- the title is confusing: "Evolutionary Training of Sparse Artificial Neural Networks: A Network Science Perspective." Within ML, evolutionary training generally refers to evolutionary algorithms that instantiate Darwinian evolution (and there is indeed a community that studies evolving neural networks through evolutionary algorithms). The proposed method has no heritable variation even though the abstract claims that the method follows a Darwinian evolutionary approach. I don't think a reader benefits from "evolutionary" in the title. Maybe there's some more descriptive title, e.g. "Training Artificial Neural Networks with Adaptive Sparse Connectivity." At minimum, the claim that the method instantiates Darwinian evolution should be dropped -- it does not. Further, network science seems to have a mild influence here -- mostly the idea is impose sparsity, and while the training does result in a scale-free structure -- how does that help us understand anything -- is that the reason why it works? If so, then there should be a control that shows a similar approach without 'scale-free' structure does not work well.

So overall, I think this is interesting work -- but the claims need to be scaled back or better supported, the term "evolutionary" needs to be clarified, and the importance of network science should be made more explicit, if it is to remain as a part of the title.

Reviewer #1:

Probably the most important idea behind this paper is that neural networks can be designed as sparse scale-free networks, and that this does not necessarily hurt what can be learned, that learning can be faster, and that this can have a positive impact on generalization. All of these things are important.

We thank the reviewer for appreciating our ideas and the potential of our proposed method.

The paper has weaknesses however. The sparseness ideas advanced by the paper (by itself) are fine. But what the authors is doing is not “Evolutionary Training” and is at best a very crude kind of local search (local random sampling is probably a better description). In addition to sparseness, the authors are removing edges with weights that are close to zero, and also adding in new random weights. A more sophisticated version of this idea (albeit one-sided) is the “Weight Elimination” methods introduced by Wiegand/Rumelhart/Huberman (NIPS 1991), which used gradient methods to remove edges with weights near zero. There have also been more complex methods using genetic algorithms to do this sort of thing, but much of this work was also in the 1990s and at the time, was only applied to small problems. The problem with this early work was scalability, but this work is getting attention again and the scalability issues have changed (obviously). What is being done here is potentially important, but it is being done in a very simplistic manner.

We do agree with the reviewer that our method is simple. It was in our scope to create a simple, but at the same time a very efficient method to address the scalability issues of artificial neural networks. In general, the simple and efficient solutions of hard problems have a high potential of being adopted by the community (e.g. Dropout). As our proposed method fulfills both criteria (clarified on page 5, Discussion section, 1st paragraph) we believe that it will have impact. In the revised version, we performed a deeper analysis of the proposed method in order to understand its potential importance. Also, we have added a citation to the suggested article (Page 2, 5th paragraph). We do not exclude the hypothesis that more sophisticated approaches can identify better the unimportant connections which can be removed, but we do not consider them the goal of this paper, and we intend to address them in the future. Same, for the process of adding new connections (Page 5, Discussion section, 2nd paragraph). We hope that the reviewer understand our viewpoint.

Still, the sparseness idea is important, and it is getting more attention. (I have been working with very sparse neural networks for at least 3 years.)

Thank you. It is great to hear that more researchers are paying attention to the sparsity issue.

The current paper is a sloppy (typos, poorly constructed sentences, jargon) and the experimental data is presented as a “data dump” that most readers will probably find hard to interpret. The authors would have been much better off presenting less data, and rather provide examples that better illustrate the effectiveness of the methods. Or results could have been presented in a more compact fashion. Most of the paper is probably not readable to the average reader in its current form. The quality of the presentation is less than what I would expect from a reasonable conference publication in a good venue (e.g. NIPS or AAAI or IJCNN).

In the revised version, we have addressed as much as possible the typos and the language problems. Also, we have tried to clarify many other aspects raised by the reviewers. We believe that the readability of the manuscript has been increased.

From the abstract:

“fully connected”

We corrected it in the revised version.

I don't think there are any successful applications of reinforcement learning. Well, at least not of the TD, Q-learning flavor.

Indeed, the most successful are the ones of deep reinforcement learning. We have corrected this in the revised version.

page 1: "we have hinted a similar fact"

In the revised version we changed it to "we observed a similar fact"

page 2: "highest negative weights" surely, not what you meant.

In the revised version we changed it to "largest negative weights"

Page 2: the comments about the "selection phase of natural selection" are just not accurate.

Indeed, our comment was strong and not clear. In the revised version of the paper, we have corrected it (page 2, 5th paragraph).

On page 3, I am not sure why the authors are talking about performance in terms of "nats" instead of accuracy. (I asked around to see if anyone else had seen "nats" used in this way; no one had. Still, this may be ignorance on my part.)

Natural units, or simply "nats", are a typical measure of information entropy (Shannon entropy). The expected value of the information content has different units of information given by the base of the logarithm used. For example the units for a logarithm with base 2 are called "bits", while for a natural logarithm are called "nats". There are some previous works which used "nats" in a more or less similar manner, e.g. Salakhutdinov and Murray, 2008.

Reviewer #2:

The paper presents an evolutionary-inspired training method to create sparse networks (for Boltzmann machines and MLPs), which are then in turn trained through unsupervised/supervised training methods.

The novel contribution of the paper is to show that ANNs can in fact perform well with sparsely-connected layers on a variety of different problems. This indicates that not as many ANN parameters are actually needed as are used in most current more fully-connected networks. Interestingly, the approach could also be used to estimate the real problem dimensionality of a problem by counting the number of remaining weights.

We thank the reviewer for summarizing our main contribution and for suggesting us some future research directions.

To improve the manuscript I would suggest the following:

- The mentioned neuroevolution (NE) examples seem a little arbitrary. For example, the popular NEAT method is not mentioned. Additionally, most NE methods do evolve sparse networks but differently than the approach in this paper. It would be good to highlight the differences in this paper to the traditional use of evolution for neural networks (which involves adding nodes and connections over evolutionary time).

In the revised version of the paper, we have added a new paragraph (page 2, 2nd paragraph) to discuss in details the fundamental differences between NEAT like methods and our approach, explaining also the advantages of our proposed method.

- It would be interesting to see example of the evolved weight patterns (maybe for MNIST). Are there any regularities in these patterns? Do they depend on the domain?

Indeed, the reviewer has intuited correctly. Besides the tendency of the hidden neurons connections to follow a power-law, also the visible neurons connectivity tends to transform into a pattern which is dependent on the domain data. As the reviewer suggested this behavior may be used to find the real problem dimensionality of a problem. To illustrate this, in the revised version, we have added a new figure (Figure 6) with patterns on the MNIST and CALTECH 16x16 datasets. Also, we have discussed these patterns in a new paragraph added in the Results section, SET performance on restricted Boltzmann machines

subsection. (Page 4, 2nd paragraph). However, the research directions described in the above mentioned paragraph do not belong to the goals of this paper, and we would prefer to let them for further work.

- Ideally, it would be good to see how the approach works together with the more commonly used convolutional networks (e.g. only replacing the last fully connected layers with SET) for image-based domains. Is there still an advantage in those cases? If this was the case it would further increase the impact of the paper.

As the reviewer suggested, we performed an extra experiment on the CIFAR 10 dataset using standard CNN, and in which we replaced the last fully connected layers with SET layers. The results confirmed the findings from RBM and MLP, and show that SET-CNN achieves better accuracy than fully connected CNN, while having about 4% of its parameters. In the revised version, we have added a new paragraph in the Results section (page 5, 2nd paragraph) to discuss this experiment. Also, we added a new figure (Figure 7) to illustrate it.

Minor comments:

- Which test is used to determine the p-values? Does the test rely on normal distributions?

We used a one-tailed test to determine the p-values. We assessed the SET hidden neurons connectivity against a power-law distribution. We clarified this aspect in the revised manuscript. Also, we performed unreported experiments using a two-tailed test to assess the SET hidden neuron connectivity against a normal distribution. As expected, we observed exactly the opposite. Initially the hidden neuron connectivity was a binomial distribution and during the training process it was slowly transforming in something else (i.e. the power-law). However, we do not consider necessary to report also the latter experiment as it does not bring too much new knowledge to the paper.

- The network notation in Table 3 is never really described. I assume all layers are fully-connected so each layer is replaced by SET?

Yes, in Table 3 we have used fully connected layers for MLP, sparse layers with fixed topology for MLP_{FixProb}, and sparse evolutionary layers for SET-MLP, while keeping exactly the same the other network hyper-parameters. In the revised version, we added a clarification in the Table 3 caption.

In conclusion, an interesting paper that raises an important point about the current general practice in neural networks.

We thank the reviewer for finding interesting our work and for the feedback. We hope that he/she is pleased with the revised version of the paper.

Reviewer #3:

As listed in the abstract, the major claims of the paper are (1) that network sparsity is unappreciated within neural network training, that (2) the proposed SET method reduces parameters and computation time quadratically relative to standard training approaches while improving performance, and that (3) the method will enable billion-neuron networks that are intractable with current methods.

The paper is interesting, the approach seems promising as viewed by the results, but the claims are reaching and generally undersupported relative to their extremity. As a result, I recommend the authors revise and resubmit their paper.

We thank the reviewer for finding interesting and promising our work, for the feedback, and for giving us the chance to improve it.

For the first claim: The value of sparsity is not unknown within deep learning. Drop-out of FC is common (and is used in the experiments), and in effect trains an ensemble of sparse networks to great effect (although the level of sparsity is less than that advocated in this paper). L1 regularization is well-known and induces pressure towards sparsity -- the authors should perform a control using L1 (the authors use weight decay, which is L2 regularization, and is not motivated directly by sparsity). For more sophisticated approaches to sparsity, see <https://arxiv.org/pdf/1702.06257.pdf> and <https://arxiv.org/pdf/1412.1442.pdf>.

Indeed, the sparsity in ANNs is successful and widely used at the neurons level (e.g. dropout), but largely ignored at the connectivity level, even if recently we have been able to see a revival of this area. As the reviewer suggested us, we performed a controlled experiments to study the effect of different regularization techniques at the weights level. We have added Figure 6 in the revised version to depict these experiments. Also, we added a new paragraph in the Results section (page 4, last paragraph), at the end of the subsection with MLP experiments to discuss the effect of regularization techniques on SET. We believe that L1 did not offered the expected results on SET-MLP due to the fact that already it has a very small number of weights. If those weights are further forced to go towards zero then the model will not have enough discriminative power.

For the second claim: It is interesting that the methods can greatly reduce the number of trainable parameters (although this potential is already established as the authors note in the network compression literature), in particular that you can start training sparsely instead of starting FC and going towards sparse. However, this prior precedent should be cited: <https://arxiv.org/pdf/1606.06216.pdf>. More problematic is the claim that computation time can be similarly reduced -- this requires empirical support. GPUs are not optimized for sparse calculations, and it is not clear that the potential for computational acceleration can actually be realized (see <https://arxiv.org/pdf/1506.02515.pdf> and <https://arxiv.org/pdf/1702.06257.pdf> to see methods informed by such limitations). Timing information from Keras on GPUs should be provided if the method really does decrease computational costs quadratically as claimed.

Indeed, one of the main advantage of our work in comparison with state-of-the-art is that SET based models start with sparse topologies, while the others not. The issue that GPUs and, in general, all the deep learning libraries are optimized for dense matrix multiplications is true. Thus, we hope, that our results will change somehow this in the future. However, we put some serious thoughts on this aspect and we realized that the actual implementations of sparse matrix multiplications are not a solution. For instance, using sparse matrix multiplication in Matlab (for the RBM experiments) we obtained 3-4 times faster running time than with dense matrix multiplication. The Tensorflow sparse matrix multiplication is even worse (1-2 times faster). These values are very far from the theoretical advantage of our method. Thus, currently, we are working on a low-level implementation of SET, which consider its particularities and the state-of-the-art technology. On short, the basic idea is to still use dense matrix multiplications and to perform batch computations. The activations of the neurons from the same layer can be computed in parallel, and the input weights to each neuron can be multiplied with data batches. Both being dense matrices/tensors. There are some more alternatives on this research direction and we are exploring them right now. However, we believe that they do not constitute the goal of this paper and we prefer to let them outside of this manuscript. We have added a short phrase in the Discussion section (page 6, last paragraph) to discuss these issues and we have cited the suggested references.

For the third claim: The hard part of any method is scaling it -- raw extrapolation is nearly always deceiving. That this method really will enable billion-neuron networks where other methods would not seems like something that it would be wise to be skeptical of (given that the networks in the paper are on the order of ~10,000 nodes, several orders of magnitude shy of one billion). To claim that the method will enable solving currently-intractable tasks when it has not been applied on tasks of moderate complexity like imagenet also seems like a large stretch, where raw extrapolation is more likely than not deceiving. The ANN literature is

littered with ideas that seemed promising at small scales but did not scale.

As this manuscript is focused on the algorithmic side and shows an excellent theoretical decrease of the number of parameters which have to be optimized, we had two reasons to stay in these range of networks:(1) a clear comparison with state-of-the-art models which does not scale (e.g. fully connected models), (2) to show the wide applicability of SET in various settings and ANN models (e.g. in RBMs, MLPs, CNNs). In our current developments of SET which are focused on the implementation aspect, and as mentioned previously we would prefer to keep them outside of this manuscript, we are able to push on a standard laptop (with 16GB RAM) the size of an SET-MLP to several hundred thousands neurons (due to its low memory footprint), while the dense connected MLP can not have more than few tens thousands neurons.

I can imagine readers will find this paper interesting, as I wouldn't have anticipated that the method would work as well as it does, and it is promising that it works both for RBMs and for MLPs, and achieves good results on permutation invariant MNIST and CIFAR-10. I'm curious to see how it scales and pairs with convolution, but I do not foresee it much influencing the field at large yet -- there are many papers targeting sparsity and compression, and usually the ML field is moved by big results (for better or worse) -- i.e. state of the art in a relevant domain, which this paper lacks.

Yes, indeed the method is simple and very efficient. In the revised version, we have added a new set of experiments on Convolutional Neural Networks. We found that SET improves CNNs in the same manner as it improves RBMs or MLPs. In the revised version, we have added a new paragraph in the Results section (page 5, 2nd paragraph) to discuss this experiment and a new figure (Figure 7) to illustrate it. We really believe, that this is a big result, and we hope that the research community will receive well our work and will take it further, while on the application side, the industry will adopt it for real-world applications.

Some nit-picks -- the title is confusing: "Evolutionary Training of Sparse Artificial Neural Networks: A Network Science Perspective." Within ML, evolutionary training generally refers to evolutionary algorithms that instantiate Darwinian evolution (and there is indeed a community that studies evolving neural networks through evolutionary algorithms). The proposed method has no heritable variation even though the abstract claims that the method follows a Darwinian evolutionary approach. I don't think a reader benefits from "evolutionary" in the title. Maybe there's some more descriptive title, e.g. "Training Artificial Neural Networks with Adaptive Sparse Connectivity." At minimum, the claim that the method instantiates Darwinian evolution should be dropped -- it does not. Further, network science seems to have a mild influence here -- mostly the idea is impose sparsity, and while the training does result in a scale-free structure -- how does that help us understand anything -- is that the reason why it works? If so, then there should be a control that shows a similar approach without 'scale-free' structure does not work well.

As the reviewer suggested, we removed the Darwinian evolution claim. Also, we propose a new title, as our method is not "evolutionary" in the straight forward sense given by the pure definition from the traditional "neuroevolution" subfield of artificial intelligence. Still, we believe that our method is "evolutionary" in a more philosophical sense. E.g., a basic genom, some simple surviving rules, and some random mutations which lead to evolution. We would prefer to keep the "network science" term in the title as the tendency of the hidden neurons connectivity to become scale-free is not enforced by us, but it is a consequence of this random evolutionary process. Thus, this is similar with many real-world complex networks, including the biological neural networks (at least at the macro-scale, as we do not have the technology yet to create a graph of a complex brain at the neurons level). We hope that the reviewer agrees with us that the comparisons with RBMFixProb, MLPFixProb, and CNNTFixProb represent our controlled experiment. These show that a fixed binomial structure (i.e. not scale-free) given by the default Erdos Renyi random graphs achieves always lower performance than the structures evolved by SET (i.e. power-law for the hidden neurons, and something close to the data distribution for the visible neurons) from the original binomial structures.

So overall, I think this is interesting work -- but the claims need to be scaled back or better supported, the

term "evolutionary" needs to be clarified, and the importance of network science should be made more explicit, if it is to remain as a part of the title.

We thank the reviewer once more to find interesting our work and for giving us feedback to improve it. We tried to address it in such a way that the revised version has a better equilibrium between claims and their support. We hope that the reviewer will still find the revised version interesting, but at the same time much more clear.

Yours sincerely,

Decebal Constantin Mocanu

On behalf of the authors of the submitted paper NCOMMS-17-18792-T.

Reviewers' comments:

Reviewer #1 (Remarks to the Author):

In some ways the paper has not changed very much. The focus has shifted somewhat away from an "evolutionary perspective," and the algorithm itself is perhaps a bit clearer.

The comments about NEAT are clearly from someone who does not understand NEAT. I don't work with NEAT myself, but I have attended a tutorial and several papers on NEAT and these authors do not understand the recursive scalability of NEAT (or hyperneat). This is a side issue, but the authors didn't get it right.

The contribution is still fundamentally this: the authors do interesting experiments showing the advantage of sparse networks. How they do this is a bit ad hoc. And there is not really any mathematics to back this up. But this is still potentially important.

The papers is still surprisingly poorly written. There are many awkward sentences that are hard to read.

Reviewer #2 (Remarks to the Author):

Almost all my previous comments have been addressed in the revised manuscript. In general, I would recommend publication after addressing the minor additional points below:

In some places the distinction between SET and traditional EC approaches could still be made more clear. For example in:

"SET follows the natural simplicity of the evolutionary approaches, which were explored successfully in our previous work on evolutionary function approximation¹⁵. Also, they have been explored for

network connectivity in¹⁶, and for the layers architecture of deep neural networks¹⁷", are you saying papers 16&17 follow the SET approach? At least the paper by Miikkulainen et al. employs an approach that builds on NEAT.

Recent advances in ES (<https://arxiv.org/abs/1703.03864>), and GA (<https://arxiv.org/abs/1712.06567>) could also be mentioned, although they have not been published yet (I believe). They do however show that purely evolutionary approaches can scale to large dimensionalities in some cases.

Was the approach ever tested with removing random weights instead of the smallest ones? I agree that it makes sense to remove the smallest ones because they probably have the least impact but it could be interesting to see how much difference there is between random vs. small_weight removal.

- Minor comments:

"To avoid being trapped in the same type of scalability issues, in SET, we focus on using the best from both world" -> both worlds?

Reviewer #3 (Remarks to the Author):

The paper is much improved by the revision. I have a few minor comments, so I recommend accept with minor revisions.

The concluding sentence:

"ANNs built with SET will have much more representational

power, and better adaptive capabilities than the current state-of-the-art ANNs, and will push artificial intelligence well beyond

its current boundaries." -- Temper this sentence! Claim without evidence, based on many assumptions about what boundaries constrict AI. There is no reason why it *will* push AI beyond boundaries -- the future of AI research (and most research fields) is not deterministic.

The CNN experiments are interesting, but some details are missing. From the network description it appears as if you performed dropout on the convolutional layers but not the fully-connected layers?

From my working knowledge that it is very atypical, because drop-out is a regularization technique and the fully-connected layers are the ones in which overfitting is a danger (because of the huge number of parameters). If drop-out was included in FC that should be mentioned, and if it wasn't in the FC then either a convincing argument for why dropout isn't included in FC should be given, or experiments with dropout in the FC should be included.

The regularization experiments in fashion MNIST are informative, but the level of L1 regularization is not mentioned in the text (as far as I could tell). Also, in general you need to (obviously) tune L1 and L2 rates, so I assume that you gave it a fair shake in your experiments (and didn't just tune them to fit your SET model), but you should mention a little bit about how you selected those hyperparameters (and of course, what the L1 level was in the plots you show).

"We anticipate that our approach will enable ANNs having billions of neurons and evolved topologies to be capable of handling complex

real-world tasks that are intractable using state-of-the-art methods." -- This strikes me once again as reaching -- most researchers anticipate their methods to have great success and move the field and yet few do. It could be the case that you pursue SETs with billion-node networks, or it could be that it turns out computationally unreasonable (given your concerns about scaling with GPUs) etc or things change as you explore ANN sizes with orders of magnitude greater than those explored in the experiments. I strongly suggest changing 'anticipate' to 'hope,' because my prediction is that ultimately we won't see billion-node SET networks blowing away all previous results (although of course I could be wrong, and am open to revising my beliefs).

With these minor suggestions taken care of, I think this interesting paper with much empirical support should be accepted and published.

Response Letter

Manuscript details

Reference number: NCOMMS-17-18792A
Old title of the article: Scalable Training of Artificial Neural Networks with Adaptive Sparse Connectivity: A Network Science Perspective
New title of the article: Scalable Training of Artificial Neural Networks with Adaptive Sparse Connectivity using a Network Science Perspective
Authors: Decebal Constantin Mocanu, Elena Mocanu, Peter Stone, Phuong H. Nguyen, Madeleine Gibescu, Antonio Liotta

Content:

Paragraphs	Page
Response to Reviewer #1	1
Response to Reviewer #2	1
Response to Reviewer #3	2
Journal format requirements	4

Dear editors and reviewers,

The authors would like to thank the reviewers and the editors for their time and valuable comments on our article. Further on, each comment of the reviewers (in black) is followed by our response (in dark-blue). Finally, you can find the changes requested by the editorial board to make our manuscript complying with the journal format requirements. Also, in the revised version of the paper, we have highlighted the new added (or changed) paragraphs in dark-blue, with the exception of the small grammar corrections.

Reviewer #1:

In some ways the paper has not changed very much. The focus has shifted somewhat away from an "evolutionary perspective," and the algorithm itself is perhaps a bit clearer.

The comments about NEAT are clearly from someone who does not understand NEAT. I don't work with NEAT myself, but I have attended a tutorial and several papers on NEAT and these authors do not understand the recursive scalability of NEAT (or hyperneat). This is a side issue, but the authors didn't get it right.

In the revised version, we have clarified the comments about NEAT (page 2, 2nd paragraph).

The contribution is still fundamentally this: the authors do interesting experiments showing the advantage of sparse networks. How they do this is a bit ad hoc. And there is not really any mathematics to back this up. But this is still potentially important.

The papers is still surprisingly poorly written. There are many awkward sentences that are hard to read.

In the revised version, we have corrected and improved the text with the help of a native English speaker. We hope that the reviewer will be pleased with the revised version. Also, we thank the reviewer for finding important our work and for the feedback provided during these revisions.

Reviewer #2:

Almost all my previous comments have been addressed in the revised manuscript. In general, I would recommend publication after addressing the minor additional points below:

We would like to thank the reviewer for giving us very useful comments to improve the manuscript and for recommending it for publication. Further on, we address the last remaining points.

In some places the distinction between SET and traditional EC approaches could still be made more clear. For example in: "SET follows the natural simplicity of the evolutionary approaches, which were explored successfully in our previous work on evolutionary function approximation¹⁵. Also, they have been explored for network connectivity in¹⁶, and for the layers architecture of deep neural networks¹⁷", are you saying papers 16&17 follow the SET approach? At least the paper by Miikkulainen et al. employs an approach that builds on NEAT.

We have clarified the text in the revised version (page 2, 2nd paragraph). Papers 16&17 are evolutionary in the traditional style and they do not follow SET.

Recent advances in ES (<https://arxiv.org/abs/1703.03864>), and GA (<https://arxiv.org/abs/1712.06567>) could also be mentioned, although they have not been published yet (I believe). They do however show that purely evolutionary approaches can scale to large dimensionalities in some cases.

In the revised manuscript we mentioned the above two papers (page 2, 2nd paragraph). Indeed, we are aware of these recent advancements in ES and GA, and we are actively doing research to use pure evolutionary approaches in SET-like sparse ANNs.

Was the approach ever tested with removing random weights instead of the smallest ones? I agree that it makes sense to remove the smallest ones because they probably have the least impact but it could be interesting to see how much difference there is between random vs. small_weight removal.

We never tested until now the approach with random weights removal, for the same reason as the reviewer mentioned. However, for this revision, we became curious and we tried it on CIFAR10, using the same SET-MLP architecture as in the paper. The difference is huge. In the next plot you can see three types of SET-MLPs: 1) SET-MLP_{smallestWeightsRemoval} where we remove 30% of the smallest weights (the case considered in the paper); 2) SET-MLP_{randomWeightsRemoval} where we remove 30% of weights randomly; and 3) SET-MLP_{largestWeightsRemoval} where we remove the largest 30% weights. It can be observed that random weights removal reduces the performance to more than 35% accuracy, while the biggest weights removal makes the model not learn at all. These somehow support our claim that the smallest weights are the ones which have to be removed. We did not add this small experiment in the

manuscript because we already have 10 display items in the paper, but we have added a small paragraph in the Discussion section to mention it (page 6, 2nd paragraph).

- Minor comments:

"To avoid being trapped in the same type of scalability issues, in SET, we focus on using the best from both world" -> both worlds?

Yes, we corrected it in the revised version.

Reviewer #3:

The paper is much improved by the revision. I have a few minor comments, so I recommend accept with minor revisions.

The authors are grateful to the reviewer for the feedback provided during these revisions, and for the accept recommendation. Below we address the minor revision.

The concluding sentence: "ANNs built with SET will have much more representational power, and better adaptive capabilities than the current state-of-the-art ANNs, and will push artificial intelligence well beyond its current boundaries." -- Temper this sentence! Claim without evidence, based on many assumptions about what boundaries constrict AI. There is no reason why it *will* push AI beyond boundaries -- the future of AI research (and most research fields) is not deterministic.

We do agree with the reviewer about the unpredictable nature of the research fields. In the revised version we tempered seriously our claim (page 6, 3rd paragraph), using the following concluding sentence:

ANNs built with SET will have much more representational power, and better adaptive capabilities than the current state-of-the-art ANNs, and we hope that they will create a new research direction in artificial intelligence.

The CNN experiments are interesting, but some details are missing. From the network description it appears as if you performed dropout on the convolutional layers but not the fully-connected layers? From my working knowledge that it is very atypical, because drop-out is a regularization technique and the fully-connected layers are the ones in which overfitting is a danger (because of the huge number of parameters). If drop-out was included in FC that should be mentioned, and if it wasn't in the FC then either a convincing argument for why dropout isn't included in FC should be given, or experiments with dropout in the FC should be included.

Indeed, dropout was used also with the FC layers. In the revised version, we added a proposition in the CNN experiments section to illustrate this (page 5, 3rd paragraph).

The regularization experiments in fashion MNIST are informative, but the level of L1 regularization is not mentioned in the text (as far as I could tell). Also, in general you need to (obviously) tune L1 and L2 rates, so I assume that you gave it a fair shake in your experiments (and didn't just tune them to fit your SET model), but you should mention a little bit about how you selected those hyperparameters (and of course, what the L1 level was in the plots you show).

In the revised version we added the L1 and L2 levels and a short proposition about their fine tuning procedure (page 5, 2nd paragraph). In general, we have tried out L1 and L2 levels between 0.01 and 0.000001 on Fashion MNIST mainly with SReLU and without Nesterov momentum. For the same regularization type/level, always we found out that SET-MLP obtains a performance more or less similar with MLP, while $MLP_{FixProb}$ always performs worse. Moreover, if L1 is smaller than 0.00001 or L2 is smaller than 0.0001, we found out that all models always achieve a good accuracy (i.e. 88%-92%), while the smallest SET-MLP accuracy is bigger than the highest $MLP_{FixProb}$ accuracy. We did not consider necessary to add all these minor details in the paper as to support them we shall have a proper controlled experiment (i.e. grid search) on some more difficult datasets (e.g. CIFAR 10), while for the purposes of this paper we consider enough the performed small random search experiment. To clarify, as the reviewer can observe from all the experiments performed in this paper, we tried to give a fair shake to all models, and not to fine tune hyperparameters for each dataset particularly to fit best our proposed model.

"We anticipate that our approach will enable ANNs having billions of neurons and evolved topologies to be capable of handling complex real-world tasks that are intractable using state-of-the-art methods." -- This strikes me once again as reaching -- most researchers anticipate their methods to have great success and move the field and yet few do. It could be the case that you pursue SETs with billion-node networks, or it could be that it turns out computationally unreasonable (given your concerns about scaling with GPUs) etc or things change as you explore ANN sizes with orders of magnitude greater than those explored in the experiments. I strongly suggest changing 'anticipate' to 'hope,' because my prediction is that ultimately we won't see billion-node SET networks blowing away all previous results (although of course I could be wrong, and am open to revising my beliefs).

We are optimistic, and so believe that we will manage to build ANNs with billions of neurons. Of course, at the same time, proofs are needed to demonstrate the hypothesis and we do not have yet those proofs for networks with billions of neurons. For this reason, we understand the reviewer concerns (and advices) and we incorporated in the revised version the strong suggestion to change 'anticipate' to 'hope'. Also, we moved this phrase from the abstract in the last paragraph of the Introduction (page 2, 3rd paragraph).

With these minor suggestions taken care of, I think this interesting paper with much empirical support should

be accepted and published.

We thank you once more for your helpful suggestions.

Journal format requirements:

- a) Title (no punctuation): We changed slightly the title to eliminate punctuation. More exactly, we have replaced ":" with "using".
- b) Abstract (maximum 150 words): In the revised version, we reduced the abstract from 235 words to less than 150 words.
- c) Introduction (maximum 1000 words): In the revised version, we moved the SET method description from the last part of the Introduction section to the first part of the Results section. In this way, we reduced the Introduction from 1325 words to less than 1000 words, while keeping exactly the same text and information flow in the paper.
- d) Introduction (last paragraph summary of both the results and the conclusions): We have added this paragraph at the end of the Introduction section (page 2, 3rd paragraph). This paragraph is mainly composed by the text removed from the abstract. We let the color of this text still black.
- e) Code and data availability: In the revised version we added a code availability paragraph and we updated the data availability paragraph.

Yours sincerely,

Decebal Constantin Mocanu

On behalf of the authors of the submitted paper NCOMMS-17-18792A.

REVIEWERS' COMMENTS:

Reviewer #1 (Remarks to the Author):

I read the paper again.

The overall flow of the paper has been improved.

The paper is still largely empirical in nature.

But the results are potentially important.

Reviewer #2 (Remarks to the Author):

All my previous comments have been addressed and I recommend the paper for publication.

Reviewer #3 (Remarks to the Author):

After reviewing the changes I think the paper is now ready to be accepted.

Response Letter

Manuscript details

Reference number: NCOMMS-17-18792B
Old title of the article: Scalable Training of Artificial Neural Networks with Adaptive Sparse Connectivity using a Network Science Perspective
New title of the article: Scalable Training of Artificial Neural Networks with Adaptive Sparse Connectivity inspired by Network Science
Authors: Decebal Constantin Mocanu, Elena Mocanu, Peter Stone, Phuong H. Nguyen, Madeleine Gibescu, Antonio Liotta

Content:

Paragraphs	Page
Response to Reviewer #1	1
Response to Reviewer #2	1
Response to Reviewer #3	1

Dear editors and reviewers,

The authors would like to thank the reviewers and the editors for their time and valuable comments on our article. Further on, each comment of the reviewers (in black) is followed by our response (in dark-blue).

Reviewer #1 (Remarks to the Author): I read the paper again.

The overall flow of the paper has been improved.

The paper is still largely empirical in nature.
But the results are potentially important.

We thank the reviewer for providing us useful comments in the previous revision rounds and for finding potentially important our results.

Reviewer #2 (Remarks to the Author):

All my previous comments have been addressed and I recommend the paper for publication.

We would like to thank the reviewer for giving us very useful comments and suggestions for future research during the revisions. Also, we thank the reviewer for recommending the paper for publication.

Reviewer #3 (Remarks to the Author):

After reviewing the changes I think the paper is now ready to be accepted.

We thank the reviewer for guiding us during the revisions and for accepting the paper.

Yours sincerely,

Decebal Constantin Mocanu

On behalf of the authors of the submitted paper NCOMMS-17-18792B